1 af 13

# Abstract

It is possible to fit Bayesian statistical models whose parameters satisfy analytically intractable algebraic conditions by emb... finder inside a gradient-based sampling algorithm like Hamiltonian Monte Carlo. This technique has enabled important scien... the high computational cost of computing and differentiating large numbers of numerical solutions. We show that dynamica... Hamiltonian trajectory can improve performance. To choose a good guess we propose two heuristics: *guess-previous* reuse... and *guess-implicit* extrapolates the previous solution using implicit differentiation. We benchmark these heuristics on a rang... present a JAX-based Python package providing easy access to a performant sampler augmented with dynamic guessing.

# Introduction

If a modeller knows that some quantities jointly satisfy algebraic constraints, they may want to embed a root-finding problem... For example, biochemical reaction networks are governed by partially-known kinetic parameters and often satisfy steady-sta... finding problem has no analytical solution, so that its solution must be approximated using numerical methods.

Statistical inference for this kind of model is possible using gradient-based Markov Chain Monte Carlo algorithms like Hamil... the parameter gradients of root-finding problems can usually be found. Unfortunately, solving and differentiating root-finding... based MCMC imposes a substantial computational overhead, limiting the range of models that can practically be fit.

In this paper we propose to address this problem by dynamically updating the root-finding algorithm's starting guess as the... Hamiltonian trajectory. We propose heuristics for updating the guess and test these on a range of statistical models, showir... performance compared with the state of the art. We also present a Python package `grapevine` containing our implementati... well as benchmarks and convenience functions that allow users to easily fit their own statistical models using our algorithm...

Our Python package and the code used to perform the experiments reported in this paper are available at **https://github.**... the Python Package Index (package name "grapevine").

# Hamiltonian Monte Carlo

Hamiltonian Monte Carlo (HMC) and its variants sample $N$ parameter vectors $\theta_1, ..., \theta_N \in \mathbb{R}^k$ according to a target probab... If the algorithm works, then the statistical properties of the sample will approximately agree with the target distribution, i.e. $\sum$ function $f$.

See [1] for a full introduction to HMC. The core strategy, shared with the older Metropolis-Hastings-Rosenberg algorithm (MI... parameter vector $\theta^\dagger$ to generate a proposal vector $\theta^\star$, then accept or reject the proposal randomly, with probability dependin... generates proposals according to a Gaussian random walk, so that $\theta^\star \sim N(\theta^\dagger, \Sigma)$, HMC constructs an auxiliary dynamica... dynamics, then generates a proposal by numerically simulating the trajectory resulting from randomly perturbing this syster...

In more detail, the auxiliary dynamical system maps the parameter vector $\theta \in \mathbb{R}^k$ to a particle in $k$-dimensional space with... kinetic energy $K(\theta, \kappa) = -\ln \pi(\kappa \mid \theta)$ for auxiliary momentum vector $\kappa \in \mathbb{R}^k$. Perturbations are modelled by choosing a... trajectory where the Hamiltonian $H(\theta, \kappa) = K(\theta, \kappa) + V(\theta)$ is constant.

See [15] [16] for discussion of previous implementations of HMC with embedded root-finding.

While many numerical root-finding algorithms exist, below we focus on the Newton-Raphson algorithm [17]. To find $x$ satifyi
iteratively updates a starting guess $x_0$ according to the rule

$$x_{i+1} = x_i - J_{x_i}^{-1} g(x_i, \theta)$$

where $J_{x_i} = \frac{\partial g(x_i, \theta)}{\partial x_i}$ is the jacobian of the target function with respect to $x_i$.

Once a solution $x$ is found that satisfies $g(x, \theta) = 0$ to within a desired tolerance, its gradients with respect to $\theta$ can be fou

$$\frac{\partial x}{\partial \theta} = -J_x^{-1} J_\theta$$

Assuming that calculating and inverting the jacobian term $J_{x_i}^{-1}$ is approximately equally costly throughout, the computation
algorithm to solve an embedded root-finding problem for a single simulated segment of a Hamiltonian trajectory is approxim
of Newton steps required. Like other numerical root-finding algorithms, this cost is highly sensitive to the starting guess $x_0$:
solution then the algorithm converges quadratically, whereas a poor starting guess can prevent convergence altogether.
See [18] for a general discussion of the effect of the choice of starting guess on numerical root-finding performance.

A natural way to speed up HMC with embedded root-finding is to find the best possible guess for each problem. The main re
[10] is to hard-code a guess that is reasonable, given the likely values of the parameters. However, previous implementations
require the starting guess to be the same for all problems on the same simulated Hamiltonian trajectory. Since the sampling
ability to propose a parameter vector $\theta^\star$ that is distant in parameter space from the current vector $\theta^\dagger$, this limitation is proble
$\theta^\dagger$ to a distant $\theta^\star$ will include intermediate parameter vectors $\theta^a$ and $\theta^b$ that are also distant. The solution to problem $a$, i.e. 
from that of problem $b$, $g(x, \theta^b) = \bar{0}$. Therefore a starting guess that is optimal for problem $a$ will be sub-optimal for proble

## Related work

Our approach draws on previous work on numerical continuation and warm starting. Continuation refers to methods that us
an initial guess for a perturbed problem, which can then be solved more precisely using another method. In particular, our *gu*
Euler-Newton predictor-corrector method investigated by Allgower and Georg [19]. Chapter 5 of this work proves that a Euler
converges to the correct solution path of a continuously varying root-finding problem under sufficiently small linear perturba

In model predictive control it is often useful to "warm start" a numerical solver using a solution from a previous time step [20
combined in the context of bilevel optimisation of neural network hyperparameters [21]. Sambharya et al [22] propose using 
relationship between parameter values and optimal warm starts of numerical fixed-point optimisers and compare this learn
"nearest-neighbour" warm start that is similar to our method. The learned warm start performs better for problems where th
its nearest neighbour, so that the neighbour's parameter and root combination and their gradients are uninformative. In cont
on adjacent steps of a simulated Hamiltonian trajectory are usually close, and very many warm starts are required for a full 
consideration of cheaper linear warm starting procedures.

## Methods

$$\text{guess-previous}(x^{i-1}) = x^{i-1}$$

The information for the heuristic *guess-implicit* is the solution $x^{i-1}$ of the previous root-finding problem, the current parame
parameter vector $\theta^{i-1}$. The heuristic is to use implicit differentiation to find the local derivative of the previous solution with
then perturb the previous solution by the product of this derivative and the change in parameter values:

$$\text{guess-implicit}(x^{i-1}, \theta^i, \theta^{i-1}) = x^{i-1} - \frac{\partial x^{i-1}}{\partial \theta^{i-1}}(\theta^i - \theta^{i-1})$$

See Appendix 3 for details of how we implemented *guess-implicit.*

A limitation of *guess-previous* and *guess-implicit* is that they rely on a smooth relationship between the parameter vector $\theta$ a
vector $x$, so that the solution of the root-finding problem $g(x^i, \theta^i) = \bar{0}$, is informative as to the solution of the problem at th
$g(x^{i+1}, \theta^{i+1}) = 0$.

In the context of Hamiltonian Monte Carlo we hypothesise that this kind of smoothness is likely to obtain. Stable HMC samp
sufficiently smooth for a leapfrog integrator's discretisation error to remain bounded [23]. If the integrator's step size is too la
smooth, the leapfrog trajectory diverges from the true Hamiltonian flow. Such divergences are a key diagnostic tool: see [1] .
divergences and tune the integrator specifically to avoid them, for example by adjusting the step size, applying geometric tra
errors. This process naturally restricts the sampler to regimes where $\pi(\theta)$ behaves smoothly.

In problems with embedded root-finding, $\pi(\theta)$ is invariably coupled with the root $x$ of the embedded root-finding problem $g($
$x$, there would be little reason to calculate $x$ at each leapfrog step. Consequently, we hypothesise, the required smoothness
smooth relationship between the parameter vector $\theta$ and the roots $x$. This argument is difficult to make rigorous due to the
we relied on empirical tests to see whether it tends to hold in practice.

# Illustrative example

To understand the motivation behind our method, consider the illustrative example shown below in figure 1. The example sh
finding problems embedded on a Hamiltonian trajectory extracted from a representative model run, as well as the path throu
to solution taken by Newton solvers using different guessing heuristics. In this example solving the 11 embedded problems
Newton steps, whereas with *guess-implicit* only 14 Newton steps were needed.

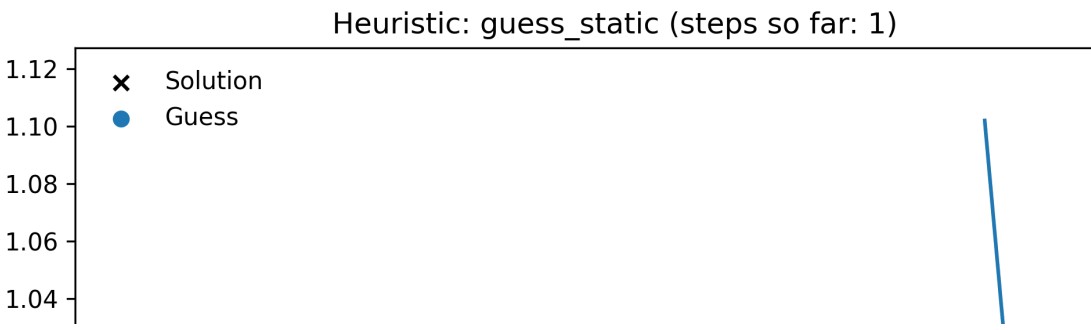

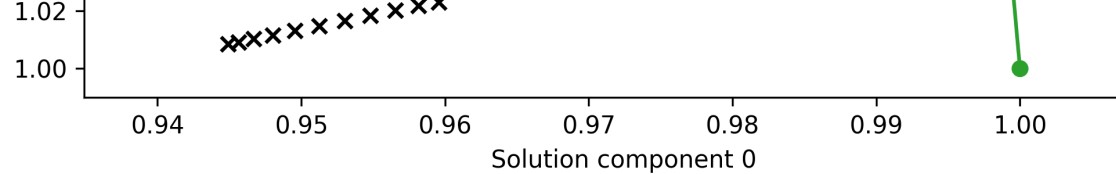

1.02

1.00

0.94       0.95       0.96       0.97       0.98       0.99       1.00

Solution component 0

Figure 1: **Illustrative example** This figure compares the behaviour of dynamic and static heuristics along a single Hamiltonian trajectory. Black c
problems (finding the minimum of a parametrised 2-dimensional Rosenbrock function) corresponding to points along a simulated Hamiltonian
lines show the paths through solution space taken by a Newton solver to approximately solve each problem. Note that some of these paths are
used for the current problem. For the first problem in the trajectory all heuristics use the default guess at coordinate (1, 1); for subsequent pro
default guess whereas the dynamic heuristics use better guesses that lie closer to the target, thus saving Newton steps. The heuristic *guess-*
gradient information.

# Performance benchmarks

Our illustrative example is encouraging, as it is roughly representative of embedded root-finding in practice, but it is not suffic
efficacy. The example only shows a single trajectory, whereas a typical MCMC run may require thousands of trajectories, wit
from one trajectory to another. The example also only considers one root-finding problem, whereas a good heuristic should p
problems. For a more comprehensive performance comparison, we compared the performance of dynamic and static gues
models with different embedded root-finding problems. These models, described in detail in Appendix 5, fall into three categ

First, there were seven simple statistical models embedding standard problems used to test optimisation algorithms, reform
problems: "Easom", "Beale", "Rastrigin (3d)", "Rosenbrock (3d)", "Rosenbrock (8d)", "Styblinski Tang (3d)" and "Levy (3d)". Thes
that dynamic guessing would solver performance and robustness on especially difficult embedded problems. We chose the
they have a range of different difficult features for numerical solvers, vary in dimensions, have global minima so that the ass
posed and are straightforward to implement.

Second, we tested two steady state metabolic network models, one relatively small ("Linear network") and one large ("Methio
common in many fields, especially biochemistry: see for example [24] [25] [12]. These models therefore served to test wheth
practically significant benefits in real applications

Third, to test our hypothesis that HMC adaptation would tend to induce a smooth relationship between the solution vector $x$
two models embedding the root-finding problem $g(x, \theta) = x^3 - x \odot \sin(k\theta) \odot \cos(k\theta)$, with $\odot$ representing element-w
parameter $k$ set to the very high value $1e8$. The solution to this problem depends non-smoothly on $\theta$ due to the oscillations
In one model, "Adversarial Dependent", the target density $\pi(x, \theta)$ depends on the solution $x$ via a Gaussian likelihood functi
application. The other model "Adversarial Independent", was exactly the same, but with $\pi$ made independent of $x$ by setting
expected that, if our hypothesis were correct, then dynamic guessing would outperform static guessing on the dependent m

For each model, we randomly generated 20 parametrisations based on the prior distribution. For each parametrisation we ra
consistently with the model's likelihood function, resulting in random draws from the models' prior predictive distributions. W
measurement set's posterior distribution using the No-U-Turn sampler, using the Stan adaptation algorithm provided by blac
the performance of the trivial heuristic *guess-static* with our proposed heuristics *guess-previous* and *guess-implicit*. Sample
were set per model, and so were the same for all heuristics. In addition, for each measurement set, we initialised each heuris
with the same random seed, so that they explored the same path through parameter space. Appendix 4 below describes our

We quantified performance using the total wall time required to complete the whole MCMC run, including warmup and samp
number of post-warmup Newton steps taken in each run. The wall time metric gives a practical indication of the benefit of o
generalisability beyond our software implementation and hardware setup. The total number of post-warmup Newton steps

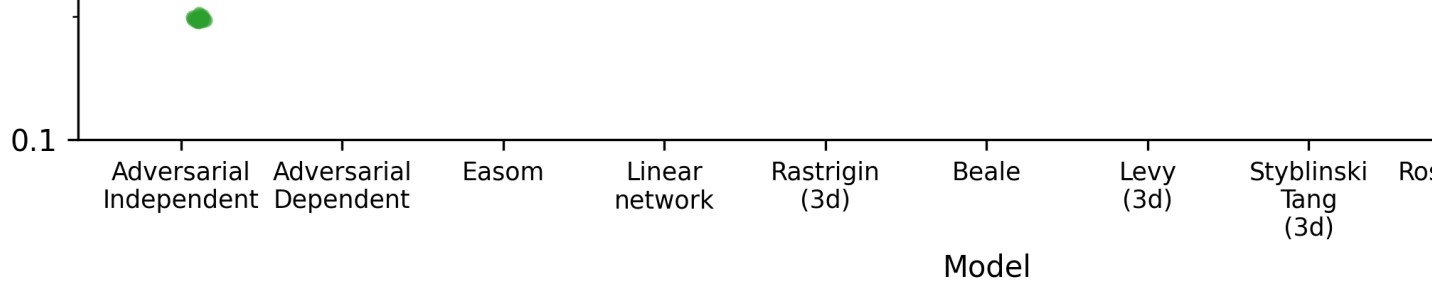

0.1

Adversarial   Adversarial              Easom          Linear        Rastrigin        Beale        Levy      Styblinski   Ros
Independent  Dependent                     network       (3d)                  (3d)       Tang
                                                                    (3d)

Model

**Figure 2: Newton step performance comparison** The relative performance of the dynamic guessing methods compared with *guess-static*. Each taken by an MCMC sampler to generate 500 post-warmup samples with the *guess-static* heuristic, divided by the number of steps taken to ger dynamic guessing. 20 random parametrisations of each model were tested.

**Figure 3: Wall time performance comparison** The relative wall-time performance of the dynamic guessing methods compared with *guess-static*. 500 samples by a sampler using *guess-static*, divided by the time taken to generate the same post-warmup samples using dynamic guessing. were tested. Each run was performed in duplicate without recomilation of code, with only the time to complete the second run recorded, so th times.

The results of our benchmarks are tabulated below in supplementary tables S1 and S2.

The number of failed MCMC runs for each heuristic and model was as follows:

| model | *guess-static* | *guess-previous* | *guess-implicit* |
|---|---|---|---|
| Rosenbrock (8d) | 19 | 2 | 2 |

All of our experiments were performed on a MacBook Pro 2024 with Apple M4 Pro processor and 48GB RAM, running macO
our benchmarks can be found in our code repository in the file `readme.md`.

## Discussion and conclusions

---

The dynamic algorithms' lower failure rate is likely because, for these algorithms, the sampler diverged less often at the star
when the sampler must simulate trajectories that traverse low-probability regions of parameter space. Such trajectories are
guessing because they have root-finding problem solutions that are far away from any reasonable global guess.

Based on our results, we expect that replacing a static guessing algorithm with dynamic guessing will typically improve sam
embedded root-finding, making it possible to fit previously infeasible statistical models. The performance improvements on
functions show that dynamic guessing can improve MCMC performance even when the embedded problem is challenging f
steady state metabolic network models show that this performance benefit extends to realistic cases, and can lead to a prac
time.

As we expected, dynamic guessing was unhelpful in the "Adversarial Independent" model, which embedded a root-finding pr
smoothly with the parameter vector, and where there was no coupling of the root-finding problem with the target probability
showed similar performance as *guess-static*, whereas *guess-implicit* performed worse due to the local parameter gradient o
solver away from the next solution. However, when we coupled the root-finding problem with the target probability density in
again saw improved performance from dynamic compared with static guessing. We conclude that HMC adaptation algorith
even highly non-smooth parameter-root relationships, provided that the roots are coupled with the target density function via

In our experiments *guess-implicit* consistently saved Newton steps compared with *guess-previous*, except in the non-repres
case. This improvement in theoretical performance mostly translated to improved wall-time on our hardware and with our so
"Linear network" and "Adversarial Dependent cases", where the *guess-previous* sampler used slightly less time despite perfo
likely due to the additional computational overhead imposed by calculating and caching $\frac{\partial x^i}{\partial \theta^i}$ at each leapfrog step. While our
quantity efficiently (see appendices 3 and 4 below for details), re-calculating it at all wastes work as $\frac{\partial x^i}{\partial \theta^i}$ is already needed to
energy (see section on embedded root-finding above). It is therefore likely possible to reduce the cost of *guess-implicit* by m
leapfrog integrator. In the meantime, we recommend using *guess-implicit* in preference to *guess-previous* because it will ten
scenario where the cost per Newton step is very high.

Our approach is not strictly limited to Hamiltonian Monte Carlo, and would also work for MCMC algorithms such as Metropo
Monte Carlo [26] that do not generate proposals by simulating continuous trajectories of an adjoint dynamical system. Howe
would only yield limited benefits for these algorithms. First, algorithms that do not require leapfrog integration, and where th
to the current proposal, can implement warm starting far more simply. Second, whereas leapfrog integration within HMC yie
densities and, we argue above, a smooth parameter-root relationship, this is not the case for other algorithms. As a result, no
such as the one proposed in [22] may work better for non-HMC algorithms than our heuristics.

An opportunity for further performance improvement would be to use a different method to solve the first root-finding proble
problems. Plausibly, a slow but robust solver could be preferable for the first problem, which uses a default guess, whereas a
preferable for later problems where a potentially better guess is available.

There are several ways in which our approach can fail, beyond those explored in our experiments. For embedded problems v
solution may be a bad guess for the next solution. If the jacobian $J_x$ is singular or near-singular then our conjugate gradient
the target probability distribution has varying characteristic length scale, so that a too-large leapfrog integrator step size is c
a suboptimal starting guess. These failure modes are shared with non-linear root-finding and Hamiltonian Monte Carlo in ge
exacerbate them.

| | | | |
|---|---|---|---|
| Styblinski Tang (3d) | 12742 (↓10557, ↑15834) | 8959 (↓8038, ↑9517) | **4556** (↓4097, ↑51… |
| Levy (3d) | 12450 (↓10387, ↑17051) | 10460 (↓9557, ↑12248) | **4479** (↓4072, ↑50… |
| Rosenbrock (8d) | 18202 (↓18202, ↑18202) | 15868 (↓15210, ↑16975) | **5691** (↓5504, ↑59… |

Table S1: **Newton step performance comparison** Average, minimum (↓) and maximum (↑) Newton step counts for each heuristic and n…

| model | *guess-static* | *guess-previous* | *guess-implicit* | |------------------------|------------------------------------|-------------------------------… Rosenbrock (3d) | 0.55 (↓0.41, ↑0.87) | 0.40 (↓ 0.37, ↑ 0.45) | **0.30** (↓ 0.27, ↑ 0.36) | | Easom | 0.25 (↓0.23, ↑0.26) | 0.24 (↓ 0… Beale | 0.29 (↓0.27, ↑0.31) | 0.28 (↓ 0.26, ↑ 0.31) | **0.24** (↓ 0.23, ↑ 0.26) | | Styblinski Tang (3d) | 0.36 (↓0.31, ↑0.42) | 0.29 (↓ … Adversarial Independent | **0.04** (↓0.04, ↑0.05) | 0.04 (↓ 0.04, ↑ 0.05) | 0.14 (↓ 0.13, ↑ 0.15) | | Methionine cycle | 1218.53 (… 124.10) | **74.16** (↓ 63.87, ↑ 159.66) | | Linear network | 0.37 (↓0.35, ↑0.40) | **0.31** (↓ 0.30, ↑ 0.34) | 0.34 (↓ 0.33, ↑ 0.36) | … 0.54 (↓ 0.52, ↑ 0.60) | **0.44** (↓ 0.41, ↑ 0.50) | | Adversarial Dependent | 0.07 (↓0.05, ↑0.07) | **0.06** (↓ 0.05, ↑ 0.07) | 0.07 (↓ … ↑0.46) | 0.35 (↓ 0.34, ↑ 0.43) | **0.31** (↓ 0.29, ↑ 0.37) | | Rastrigin (3d) | 0.31 (↓0.30, ↑0.33) | 0.30 (↓ 0.29, ↑ 0.33) …

Table S2: **Wall time performance comparison** Average, minimum (↓) and maximum (↑) time in seconds to generate 500 post-warmup samples, f… not included. The time includes both warmup and sampling phases. Each run was performed in duplicate without recomilation of code, with the… recorded results exclude compilation times.

# Appendix 2: pseudocode description of dynamic g

We assume we have:

- a function $\mathrm{Solve}$ that takes in a guess and returns an approximate solution to the target root-finding problem

- a scalar-valued function $\mathrm{LogProb}$ that takes in a set of parameter values and a root-finding solution

- a function $\mathrm{GetInfo}$ that takes in a set of parameter values and a root-finding solution, returning information for a heuris…

- a function $\mathrm{Heuristic}$ that takes in the output of $\mathrm{GetInfo}$ and returns a guess

- initial parameters $\theta_0$

- initial momentum $\kappa_0$

- default information $\mathrm{info}_{default}$

- step size $\epsilon$

The aim is to initialise and then simulate a Hamiltonian trajectory using Leapfrog integration, while passing root-finding infor… the functions $\mathrm{Heuristic}$, $\mathrm{LogDensityAndInfo}$, $\mathrm{PotentialGradientAndInfo}$, $\mathrm{InitialiseTrajectory}$ and $\mathrm{LeapfrogSte…}$

**Function 1: Generate a new guess ($\mathrm{Heuristic}$)**

- **Input:**
  - Parameters $\theta$

  - Information from previous step, $\mathrm{info}$

- **Output:** A guess for the root-finding problem, $x_0$

For example, for the heuristic *guess-previous*, $\mathrm{info}$ is the solution from previous step, if available, or a dummy value indicatin… trajectory. The output is the previous solution, if available, or else a default value.

4. Return $V$, $\nabla_\theta V$ and $\mathrm{info}_+$

**Algorithm 4: Initialise Trajectory ($\mathrm{InitialiseTrajectory}$)**

1. $V_0, \nabla_\theta V_0, \mathrm{info} \leftarrow \mathrm{PotentialGradientAndInfo}(\theta_0, \mathrm{info}_{default})$

2. Return $\theta_0, \kappa_0, V_0, \nabla_\theta V_0, \mathrm{info}$

**Algorithm 5: Leapfrog Integration Step ($\mathrm{LeapfrogStep}$)**

- **Input:**
  - Current state $\theta, \kappa, V, \nabla_\theta V, \mathrm{info}$

  - step size $\epsilon$

1. $\kappa_{\mathrm{mo}} \leftarrow \kappa - \frac{\epsilon}{2}\nabla_\theta V$ (Update momentum, first half-step)

2. $\theta_+ \leftarrow \theta + \epsilon\kappa_{\mathrm{mo}}$ (Update parameters)

3. $V_+, \nabla_\theta V_+, \mathrm{info}_+ \leftarrow \mathrm{PotentialGradientAndInfo}(\theta_+, \mathrm{info})$

4. $\kappa_+ \leftarrow \kappa_{\mathrm{mo}} - \frac{\epsilon}{2}\nabla_\theta V_+$ (Update momentum, second half-step)

5. Return $\theta_+, \kappa_+, V_+, \nabla_\theta V_+, \mathrm{info}_+$

The functions $\mathrm{InitialiseTrajectory}$ and $\mathrm{LeapfrogStep}$ are drop-in substitutes for their counterparts in standard Hamiltor

# Appendix 3: implementation of *guess-implicit*

The *guess-implicit* heuristic is defined as follows, given previous solution $x^{i-1}$, previous parameters $\theta^{i-1}$ and current param

$$\mathrm{guess\text{-}implicit}(x^{i-1}, \theta^{i-1}, \theta^i) = x^{i-1} + \frac{dx}{d\theta}\Delta\theta$$

where $\Delta\theta = (\theta^i - \theta^{i-1})$.

To obtain $\frac{dx}{d\theta}$, we use the following consequence of the implicit function theorem [27]:

$$\frac{\partial x}{\partial \theta} = -(\mathrm{jac}_x\, g(x^{i-1}, \theta^{i-1}))^{-1}\, \mathrm{jac}_\theta\, g(x^{i-1}, \theta^{i-1})$$

In this expression the term

$$\mathrm{jac}_x\, g(x^{i-1}, \theta^{i-1})$$

abbreviated below to $J_x$, indicates the jacobian with respect to $x$ of $g(x^{i-1}, \theta^{i-1})$. Similarly

$$\mathrm{jac}_\theta\, g(x^{i-1}, \theta^{i-1}) = J_\theta$$

to avoid materialising the matrix $J_x$ using a similar strategy, as demonstrated by the function `guess_implicit_cg` below.

```python
import jax

def guess_implicit_cg(guess_info, params, f):
    "Guess the next solution using the implicit function theorem."
    old_x, old_p, *_ = guess_info
    delta_p = jax.tree.map(lambda o, n: n - o, old_p, params)
    _, jvpp = jax.jvp(lambda p: f(old_x, p), (old_p,), (delta_p,))

    def matvec(v):
        "Compute Jx @ v"
        return jax.jvp(lambda x: f(x, old_p), (old_x,), (v,))[1]

    dx = -jax.scipy.sparse.linalg.cg(matvec, jvpp)[0]
    return old_x + dx
```

Which implementation of *guess-implicit* is preferable depends on the relative cost and reliability of directly calculating the m
`guess_implicit`, compared with numerically solving $J_x J_p \nabla_p = 0$ as in `guess_implicit_cg`. In general, this depends on th
the numerical solver relative to direct matrix inversion as implemented by the function `jax.numpy.linalg.inv`. For example
definite, `guess_implicit_cg` will likely perform better as the conjugate gradient method can exploit sparsity[28].

# Appendix 4: software details

Using Blackjax[7], we implemented a version of a No-U-Turn sampler with dynamic guessing, which we call "grapeNUTS". Fo
package `grapevine` containing our implementation, including a utility function `run_grapenuts` with which users can easily t

Our implementation builds on the popular JAX[29] scientific computing ecosystem, allowing users to straightforwardly defi
models to work with grapeNUTS. Similarly to Bayeux[30], grapevine requires a model in the form of a function that returns a
JAX PyTree of parameters; additionally, in grapevine such a function must also accept and return a PyTree containing inform
embedded root-finding problems. Users can specify root-finding problems using arbitrary JAX-compatible libraries, for exam

# Appendix 5: benchmark models

## Optimisation test functions

We compared our four heuristics on a series of variations of the following model:

$$\theta \sim Normal(0, \sigma_\theta)$$
$$x \sim Normal(\hat{x}, \sigma_x)$$

where $\hat{x}$ is the root such that $f(\hat{x} + \theta) = 0$.

In this equation $f$ is the gradient of a textbook optimisation test function, $sol$ is the textbook solution and $\theta$ is a vector with
tested the following functions from the virtual library of simulation experiments[32]:

The model "Adversarial-Dependent" was the same as the test function models, i.e.

$$\theta \sim Normal(0, \sigma_\theta)$$
$$x \sim Normal(\hat{x}, \sigma_x)$$

In this model the solution $x$ is coupled with the total log probability density $\pi(\theta)$ by the likelihood .

The model "Adversarial Independent" was the same, but without the likelihood, i.e.

$$\theta \sim Normal(0, \sigma_\theta)$$

The code implementing "Adversarial Independent" nonetheless evaluated $x$ at every leapfrog step.

## Steady-state reaction networks

To illustrate our algorithm's practical relevance we constructed two statistical models where evaluating the likelihood $p(y \mid$
problem, i.e. finding a vector $x$ such that $\frac{dx}{dt} = S \cdot v(x, \theta) = \bar{0}$ for known real-valued matrix $S$ and function $v$. In the contex
$S_{ij} \in \mathbb{R}$ can be interpreted as representing the amount of compound $i$ consumed or produced by reaction $j$, $x$ as the abun
as the rate of each reaction. The condition $\frac{dx}{dt} = \bar{0}$ then represents the assumption that the compounds' abundances are co

We tested two similar models with this broad structure, one embedding a small biologically-inspired steady state problem ar
studied realistic steady-state problem.

The smaller modelled network is a toy model of a linear pathway with three reversible reactions with rates $v_1$, $v_2$ and $v_3$. The
concentrations $x_A$ and $x_B$ according to the following graph:

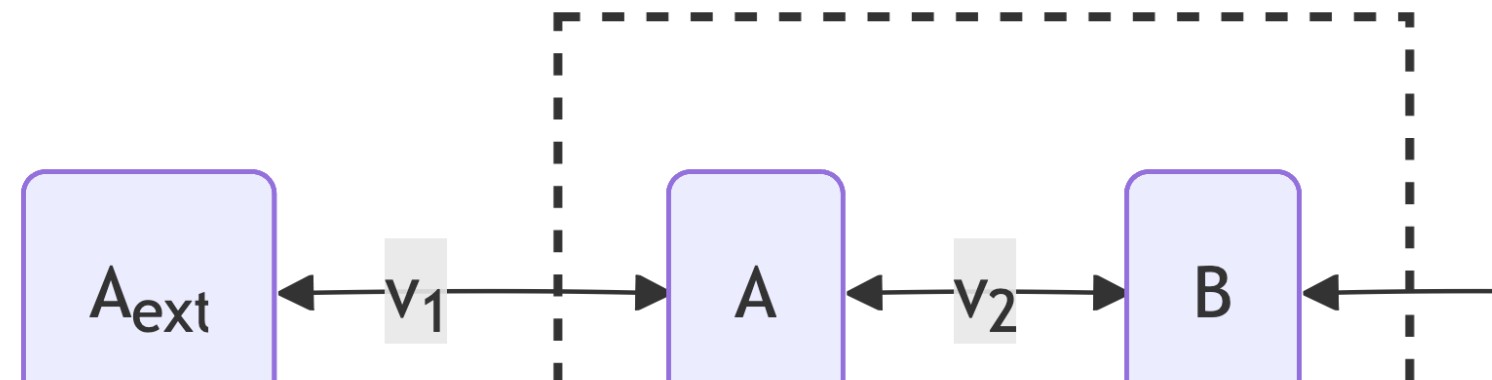

are calculated as follows:

$$v_1(x, \theta) = k_1^f(x_A^{ext} - x_A/k_1^{eq})$$

$$v_2(x, \theta) = \frac{\frac{v^{max}}{k_A^m}(x_A - x_B/k_2^{eq})}{1 + x_A/k_A^m + x_B/k_B^m}$$

$$v_3(x, \theta) = k_3^f(x_B^{ext} - x_B/k_3^{eq})$$

According to these equations, rates $v_1$ and $v_3$ are described by mass-action rate laws: transport reactions are often modelle the Michaelis-Menten equation that is a popular choice for modelling the rates of enzyme-catalysed reactions.

The larger network models the mammalian methionine cycle, using equations taken from [12], including highly non-linear reg model because it describes a real biological system and has a convenient scale, being large and complex enough to test the small enough for benchmarking purposes.

For the small linear network, we solved the embedded steady-state problem using the optimistix Newton solver. For the larg simulated the evolution of internal concentrations as an initial value problem until a steady-state event occurred, using the s ODE solver provided by diffrax. In this case a guess is still needed in order to provide an initial value. Solving a steady-state p than directly solving the system of algebraic equations; see [25] and [33] for further discussion.

Code used for these two experiments is in the code repository files `benchmarks/methionine.py` and `benchmarks/linear.p`

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

BibTeX citation

```
@inproceedings{groves2026dynamicguessingfor,
    author = {Groves, Teddy and Cowie, Nicholas Luke and Nielsen, Lars Keld},
    title = {Dynamic guessing for Hamiltonian Monte Carlo with embedded numerical root-fir
    booktitle = {TMLR Beyond PDF},
    year = {}
}
```

22. **Learning to Warm-Start Fixed-Point Optimization Algorithms**   [link]
    Sambharya, R., Hall, G., Amos, B. and Stellato, B., 2024. J. Mach. Learn. Res., Vol 25(1), pp. 166:7854--166:7899.

23. **Geometric Integrators and the Hamiltonian Monte Carlo Method**   [link]
    Bou-Rabee, N. and Sanz-Serna, J.M., 2018. Acta Numerica, Vol 27, pp. 113--206. DOI: 10.1017/S0962492917000101

24. **GRASP: A Computational Platform for Building Kinetic Models of Cellular Metabolism**   [link]
    Matos, M.R.A., Saa, P.A., Cowie, N., Volkova, S., de Leeuw, M. and Nielsen, L.K., 2022. Bioinformatics Advances, Vol 2(1), pp. vbac066. DOI: https://doi.org,

25. **Tailored Parameter Optimization Methods for Ordinary Differential Equation Models with Steady-State Constraints**   [link]
    Fiedler, A., Raeth, S., Theis, F.J., Hausser, A. and Hasenauer, J., 2016. BMC Systems Biology, Vol 10(1), pp. 80. DOI: https://doi.org/10.1186/s12918-016-0

26. **Fluctuation without Dissipation: Microcanonical Langevin Monte Carlo**   [HTML]
    Robnik, J. and Seljak, U., 2024. Proceedings of the 6th Symposium on Advances in Approximate Bayesian Inference, pp. 111--126. PMLR.

27. **The Implicit and the Inverse Function Theorems: Easy Proofs**   [PDF]
    de Oliveira, O.R.B., 2014. Real Analysis Exchange, Vol 39(1), pp. 207. DOI: https://doi.org/10.14321/realanalexch.39.1.0207

28. **Lecture Notes: Optimization III**   [PDF]
    Ben-Tal, A. and Nemirovski, A., 2023.

29. **JAX: Composable Transformations of Python+NumPy Programs**   [link]
    Bradbury, J., Frostig, R., Hawkins, P., Johnson, M.J., Leary, C., Maclaurin, D., Necula, G., Paszke, A., VanderPlas, J., Wanderman-Milne, S. and Zhang, Q., 20

30. **Bayeux: State of the Art Inference for Your Bayesian Models**   [link]
    developers, B., 2025.

31. **On Neural Differential Equations**   [PDF]
    Kidger, P., 2021. DOI: https://doi.org/10.48550/arXiv.2202.02435

32. **Virtual Library of Simulation Experiments: Test Functions and Datasets**   [link]
    Surjanovic, S. and Bingham, D..

33. **Efficient Computation of Adjoint Sensitivities at Steady-State in ODE Models of Biochemical Reaction Networks**   [link]
    Lakrisenko, P., Stapor, P., Grein, S., Paszkowski, L., Pathirana, D., Frohlich, F., Lines, G.T., Weindl, D. and Hasenauer, J., 2023. PLOS Computational Biology, V
    Science. DOI: https://doi.org/10.1371/journal.pcbi.1010783

