# OpenReview forum: "Dynamic guessing for Hamiltonian Monte Carlo with embedded numerical root-finding"
_TMLR — Accepted by TMLR_

### Review · Reviewer_qAvm · 2026-04-05

**Summary Of Contributions:**

The paper proposes a warm-starting method for HMC that involves an embedded numerical root finder to compute the potential function. The authors introduce two heuristics: _guess-previous_, which initialises the current root finding problem with the previous root solution, and _guess-implicit_, which adds a first order correction to the previous solution to warm start. Examples on several benchmark datasets show that the warm-starting method, especially _guess-implicit_ improves the efficiency of root finder-embedded HMC without warm-starting.

__Strengths:__
- The idea that is proposed is simple and easy to adopt
- Empirically, dynamic guessing seems to generally improve solver efficiency over static guessing.

__Weaknesses:__
- I have some major issues with the overall presentation of the paper. The lack of mathematical exposition makes it difficult to comprehend the problem that is being solved. I would have appreciated if the authors stated the problem setting (e.g. computational and statistical structure) more succinctly at the beginning of the paper, with more concrete examples provided, instead of deferring all discussions to the appendix or external references.
- The overall novelty of the work feels limited, as it studies a relatively niche problem and the solution is simply to warm-start the root finder  along the integration trajectory. Furthermore, this warm-start trick, if I understand correctly is not specific to HMC -- it is really just a trick used in the inner ODE solver used within HMC, but the problem itself is not HMC specific. This makes the framing of the paper weaker as it makes the contribution sound more HMC-specific than it actually appears to be.
- Empirical evaluations are also weak and not well-communicated. The figure in page 5 is not well-explained and hard to comprehend. I would say that simply adding a table showing the number of steps to convergence for each heuristic is a cleaner way to communicate the point. In addition, I would consider adding wall-clock times as well as smaller number of steps may not necessarily imply that it's faster, as for example _guess-implicit_ has an additional component of evaluating gradients, which surely adds to the computational cost.

**Audience:**

No

**Audience Explanation:**

The main idea introduced in this work seems quite incremental, essentially being a warm-start heuristic being applied to HMC, rather than proposing a new sampler or providing us with a better understanding of HMC. While novelty is not necessary for publication in TMLR, I would say the paper is still quite premature, lacking in rigour and having major presentation issues. For example, the paper does not state the precise mathematical set up of the problem, and empirical evidence is suggestive rather than fully convincing in my opinion.

**Broader Impact Concerns:**

There is no negative ethical implications of the work as far as I can see.

**Claims And Evidence:**

No

**Claims Explanation:**

The experiments do provide some evidence for the claim that warm-starting can reduce the number of integration steps necessary in the inner solver for HMC. However, there is no substantial depth in the experiments. For example,
- There is no rigorous characterisation of when the proposed method would work and when it would fail. For example, it is very likely not going to work well if $x$ is very sensitive to perturbations in $\theta$ (i.e. the mapping from $\theta$ to $x$ is not smooth).
- Examples are all on relatively simple problems where we can get reasonable results with standard HMC with static initial guesses, provided we run the chain long enough. It would be more interesting to see if there is a setting where this computational cost becomes a significant bottleneck and can only be solved through the proposed warm-starting procedure.
- Wall-clock times are not presented, which hides some of the other costs involved other than just the number of ODE steps. For example, _guess-implicit_ has an additional gradient evaluation procedure, which should be considered.

**Requested Changes:**

- I would suggest to make the problem formulation clearer mathematically, rather than relying on pure descriptions. The way in which the work is presented currently makes it hard to understand what problem is being solved and what exactly is new.
- Improve communication of empirical results using table comparisons, or better explain the figures.
- Wall clock times should be presented to assess the true practical benefits of the method.
- Currently, the method relies on ad hoc assumptions that $x$ depends smoothly on $\theta$, so that when we take a single leapfrog step of $\theta$, the corresponding $x$ does not vary significantly, which makes it reasonable to warm-start with the previously obtained $x$. This assumption should be made clearer and further investigation of robustness to this smoothness and when it will break could add more depth to the empirical evaluations.

---

> ### Author Response · Authors · 2026-04-22
> **Smoothness and HMC-specificity**
>
> Thanks very much for this review.
>
> **Smoothness**
>
> It is correct that we did not sufficiently justify the assumption of a smooth relationship between the root-finding solution $x$ and the parameter vector $\theta$.
>
> We have added a passage explicitly stating this assumption and justifying it within the context of Hamiltonian Monte Carlo. Our argument is that a) in order to be valid the leapfrog trajectory must be locally smooth in the target density $\pi(\theta)$ and b) $\pi(\theta)$ is almost always sensitive to the root-finding solution $x$ as otherwise there would be no need to embed the root-finding problem. Therefore the tuning that HMC already applies to ensure local smoothness in $\pi(\theta)$ will also ensure a locally smooth relationship between $\theta$ and $x$.
>
> > A limitation of *guess-previous* and *guess-implicit* is that they rely on a smooth relationship between the parameter vector $\theta$ and the embedded root-finding solution vector $x$, so that the solution of the root-finding problem $g(x^i, \theta^i) = \bar{0}$, is informative as to the solution of the problem at the next leapfrog step, i.e. $g(x^{i+1}, \theta^{i+1}) = 0$.
>
> > In the context of Hamiltonian Monte Carlo we hypothesise that this kind of smoothness is likely to obtain. Stable HMC sampling requires the target density to be sufficiently smooth for a leapfrog integrator's discretisation error to remain bounded<d-cite key="bou-rabeeGeometricIntegratorsHamiltonian2018"></d-cite>. If the integrator's step size is too large, or the target density is locally non-smooth, the leapfrog trajectory diverges from the true Hamiltonian flow. Such divergences are a key diagnostic tool: see <d-cite key="betancourtConceptualIntroductionHamiltonian2018"></d-cite>. Most implementations of HMC detect divergences and tune the integrator specifically to avoid them, for example by adjusting the step size, applying geometric transformations or raising diagnostic errors. This process naturally restricts the sampler to regimes where $\pi(\theta)$ behaves smoothly.
>
> > In problems with embedded root-finding, $\pi(\theta)$ is invariably coupled with the root $x$ of the embedded root-finding problem $g(x, \theta) = \bar{0}$. If $\pi(\theta)$ were insensitive to $x$, there would be little reason to calculate $x$ at each leapfrog step. Consequently, we hypothesise, the required smoothness in $\pi(\theta)$ will lead to a correspondingly smooth relationship between the parameter vector $\theta$ and the roots $x$. This argument is difficult to make rigorous due to the complexity of non-linear root-finding so we relied on empirical tests to see whether it tends to hold in practice.
>
> A fully rigorous version of this argument would require formal consideration of non-linear root-finding in general. This task would be difficult due to the diversity of possible root-finding problems, so we did not attempt it. However, we added two new models to our performance benchmarks that directly test our hypothesis empirically. Specifically, we tested our heuristics against two models that embed a highly non-smooth root-finding problem. One model coupled the solution with the target density in the normal way, via the likelihood, whereas in the other model $\pi(\theta)$ does not depend on $x$. As our argument would predict, our heuristics improved performance in the coupled model despite the non-smooth embedded problem, but did not improve performance for the non-coupled model.
>
> **HMC-specificity**
>
> It is true that warm starting is not only beneficial for HMC, but the smoothness argument outlined above would typically not apply for other MCMC algorithms. We are not aware of any non-HMC algorithms that construct locally smooth trajectories, and in general, MCMC algorithms aim to  move as far as possible from one proposal to the next, which would typically make warm-starting less effective.
>
> We added the following paragraph making this point to our discussion section:
>
> > Our approach is not strictly limited to Hamiltonian Monte Carlo, and would also work for MCMC algorithms such as Metropolis-Hastings-Rosenberg or Langevin Monte Carlo <d-cite key="robnikFluctuationDissipationMicrocanonical2024"></d-cite> that do not generate proposals by simulating continuous trajectories of an adjoint dynamical system. However, we expect that our approach would only yield limited benefits for these algorithms. First, algorithms that do not require leapfrog integration, and where the previous proposal tends to be close to the current proposal, can implement warm starting far more simply. Second, whereas leapfrog integration within HMC yields a series of smoothly varying target densities and, we argue above, a smooth parameter-root relationship, this is not the case for other algorithms. As a result, non-linear warm starting procedures such as the one proposed in <d-cite key="sambharyaLearningWarmstartFixedpoint2024"></d-cite> may work better for non-HMC algorithms than our heuristics.

---

> ### Author Response · Authors · 2026-04-22
> **More difficult problem and novelty**
>
> **More difficult problem**
>
> It is correct that our examples were all on problems that do not strictly require the use of a new algorithm, and including a case where MCMC sampling is computationally infeasible without dynamic guessing would certainly improve our paper.
>
> Unfortunately have not been able to include such a demonstration due to limitations of our benchmarking procedure. Our benchmarking procedure requires performing 20 MCMC runs in duplicate for each algorithm in each test model so as to explore a range of parameterisations and avoid compilation bias. We would like to improve our procedure to allow benchmarking a model that truly lies at the edge of computational feasibility but have not achieved this yet.
>
> To partially address the reviewer's concern, we re-ran our most difficult benchmark with wider measurement errors and prior distributions, making it both more computationally challenging (~20mins per run under static guessing) and more realistic.
>
> We somewhat disagree that our chosen models are simple. We intentionally tested a range of simple statistical models with difficult embedded problems in order to isolate the effect of our contribution, but we also tested some difficult steady-state metabolic network models. In the methionine case, our approach takes a realistic model run from minutes to seconds, which is practially significant given that a typical investigation requires many runs. In the model embedding an eight-dimensional Rosenbrock problem, the sampler with static guessing failed in all but two out of twenty runs, whereas the samplers with dynamic guessing succeeded far more often: in this case we believe we explored the performance boundary.
>
> We have expanded the description of the models in the main text to make it clearer why they were included.
>
> **Novelty of our contribution**
>
> The reviewer argues that the novelty of our contribution is limited because the problem we address is niche and our proposed solution is simple. Even though TMLR does not require novelty we would like to note that we somewhat disagree about this.
>
> First, while it is arguable that MCMC with embedded root-finding is niche, the researchers who occupy the niche are highly interested in performance improvements, and our contribution may expand the niche by making new applications possible.
>
> Second, while we agree that our proposed solutions are conceptually simple, in this case simple solutions are optimal because of the smoothness argument outlined above.
>
> Third, our solutions are non-trivial to implement as they couple the joint density function defining a statistical model with the leapfrog integrator, thus encompassing the whole HMC stack. Typical contributions in this area tend to be either at the modelling end, ignoring leapfrog integration as a computational detail, or at the algorithmic end, ignoring the details of the target density as a question for modellers. To pick out one detail, our implementations of the *guess-implicit* heuristic avoid materialisation of jacobians, which is not trivial to achieve. The best proof that our approach is difficult is that, although there are several actively maintained and used implementations of HMC with embedded root-finding and it is generally agreed that these are bottlenecked by solver performance, dynamic guessing has not been demonstrated before.
>
> Finally, even assuming our solutions are too simple to be considered novel, it would still be helpful for the field to provide a demonstration that dynamic guessing provides a performance benefit: our work could motivate other researchers to pursue more technically involved dynamic guessing approaches.

---

### Review · Reviewer_31C1 · 2026-04-16

**Summary Of Contributions:**

The authors propose a simple (but seemingly effective) strategy to make gradient-based Markov-Chain-Monte-Carlo-type of algorithms more efficient for numerical root-finding problems embedded in Bayesian inference schemes. Their main thesis is that when Hamiltonian Monte Carlo repeatedly solves embedded root-finding problems along a trajectory, it is wasteful to restart every solve from the same static initial guess, and one can often do better by reusing information from the previous integrator state. The authors propose a dynamic-guessing variant of HMC/NUTS, instantiate it with two heuristics, and show lower solver cost per effective sample across a set of benchmark problems, with especially significant gains for an implicit differentiation-based heuristic.
The authors provide a pseudocode version of their proposed algorithm, a Python library and a JAX based implementation of their methods.

**Additional Comments:**

As a disclosure, this paper is quite far out of my expertise. I will rely on other reviewers for final assessments especially regarding value of the work and contribution weight.

**Audience:**

Yes

**Audience Explanation:**

I believe the proposed method may be useful for a subset of people in the community making wide use of numerical solvers for the type of problems the authors suggest.

My impression is that the paper provides a (relatively) useful set of heuristics to reduce the (empirical) complexity of HMC root finding problems, but I am not convinced the contribution weight justifies a full journal manuscript version.

The contribution in terms of computational methods and provided libraries seems valuable, but I hesitate to assert a wide interest in the community, beyond a small subset of the audience running these specific experimental tools.

**Broader Impact Concerns:**

No concerns.

**Claims And Evidence:**

Yes

**Claims Explanation:**

Yes, the claims and contribution points the authors state are properly supported by empirical evidence.

**Requested Changes:**

- First, it would be really helpful if the authors included a brief section on the mathematical formulation of the problems they target, explicitly indicating at what point of the optimisation problem their method intervenes. This would also (perhaps) help the reader be convinced that the current existing methods are wasteful as the authors state.
- Can the authors provide some intuitions on the regimes where guess-previous should be preferred over guess-implicit, perhaps in terms of smoothness of the root map or the implicit Jacobian? The discussion hints at this but does not really quantify it.
- Is there any form of theoretical results, even under a specific problem class and assumptions regarding variational families etc. that would allow to derive any motivation or justification for the heuristics derived?
- Could the authors also expand the discussion on possible failure modes? Eg. multiple roots, branch switching, ill-conditioned Jacobians, poor initialisation... These points deserve a more explicit analysis if the contribution of the paper is to be purely empirical and heuristics-based.

---

> ### Author Response · Authors · 2026-04-22
> **Choice of heuristic, theoretical results, failure modes**
>
> Thanks very much for this review.
>
> We responded to the point about mathematical formulation [separately](https://openreview.net/forum?id=z4PfNDNAcN&noteId=oHfAke2UVd) as other reviewers also brought it up.
>
> **When to use guess-previous vs guess-implicit**
>
> We certainly did not sufficiently explain how to choose one or other or our proposed heuristics. We have expanded our discussion of this topic, noting that *guess-implicit* has generally better theoretical performance, though there is room to improve on our implementation.
>
> > In our experiments *guess-implicit* consistently saved Newton steps compared with *guess-previous*, except in the non-representative "Adversarial Independent" case. This improvement in theoretical performance mostly translated to improved wall-time on our hardware and with our software implementation, except in the "Linear network" and "Adversarial Dependent cases", where the *guess-previous* sampler used slightly less time despite performing additional Newton steps. This is likely due to the additional computational overhead imposed by calculating and caching $\frac{\partial x^i}{\partial \theta^i}$ at each leapfrog step. While our implementation calculates this quantity efficiently (see appendices 3 and 4 below for details), re-calculating it at all wastes work as $\frac{\partial x^i}{\partial \theta^i}$ is already needed to obtain the gradient of the potential energy (see section on embedded root-finding above). It is therefore likely possible to reduce the cost of *guess-implicit* by more closely integrating it with the leapfrog integrator. In the meantime, we recommend using *guess-implicit* in preference to *guess-previous* because it will tend to perform better in the worst-case scenario where the cost per Newton step is very high.
>
>
> **Theoretical results**
>
> We think the most relevant theoretical results are in the area of continuation, as our *guess-implicit* heuristic is a form of predictor-corrector method. We have expanded our related work section on this topic as follows:
>
> > Our approach draws on previous work on numerical continuation and warm starting. Continuation refers to methods that use one numerical solution to generate an initial guess for a perturbed problem, which can then be solved more precisely using another method. In particular, our *guess-implicit* heuristic is a case of the Euler-Newton predictor-corrector method investigated by Allgower and Georg<d-cite key="allgowerIntroductionNumericalContinuation2003"></d-cite>. Chapter 5 of this work proves that a Euler-Newton predictor-corrector method converges to the correct solution path of a continuously varying root-finding problem under sufficiently small linear perturbations.
>
>
> **Failure modes**
>
> This is another area that we did not previously discuss in enough detail. We added this paragraph to the discussion section:
>
> > There are several ways in which our approach can fail, beyond those explored in our experiments. For embedded problems with multiple roots, the previous solution may be a bad guess for the next solution. If the jacobian $J_x$ is singular or near-singular then our conjugate gradient based method for finding $\frac{\partial x}{\partial \theta}$ will fail. If the target probability distribution has varying characteristic length scale, so that a too-large leapfrog integrator step size is chosen, then *guess-implicit* may suggest a suboptimal starting guess. These failure modes are shared with non-linear root-finding and Hamiltonian Monte Carlo in general, but dynamic guessing may exacerbate them.

---

### Review · Reviewer_tf1g · 2026-04-17

**Summary Of Contributions:**

The paper proposes the following idea: update the initial guess of numerical root-finders embedded within Hamiltonian Monte Carlo trajectories to speed up sampling. The papers discusses two methods, either using the root from the previous integration step (guess-previous) or estimating it using the implicit function theorem (guess-implicit).

**Audience:**

Yes

**Audience Explanation:**

In my current opinion, the proposed approach is too narrow and comes across a specific trick but it will be useful to some people in the community.

**Claims And Evidence:**

No

**Claims Explanation:**

- The problem is well motivated but presentation of the paper requires a lot of work. It is worth giving background and write the details a bit more formally.  I understand it is the beyond pdf initiative but still writing can be improved significantly.

- I think it will be good to include the additional computational overhead of guess-implicit method in the performance metric which is currently solver steps per effective sample.

- guess-previous seems to not work on multiple benchmarks. This weakens the premise of the paper.

- The related work currently is a bit weak and misses out on important prior work. Please see below and explain the differences to approaches below:
Sambharya, Hall, Amos, Stellato (2024), Learning to Warm-Start Fixed-Point Optimization Algorithms.

**Requested Changes:**

Please see above.

---

> ### Author Response · Authors · 2026-04-22
> **Related work and failed runs**
>
> Thanks very much for reviewing our paper and for your helpful comments.
>
> We responded to the points about presentation, background and metrics [in a shared comment](https://openreview.net/forum?id=z4PfNDNAcN&noteId=oHfAke2UVd).
>
> **Related work**
>
> Thanks for directing us to a highly relevant paper about learned warm starting.
>
> We have expanded our related work section on warm starting as follows:
>
> > In model predictive control it is often useful to "warm start" a numerical solver using a solution from a previous time step <d-cite key="diehlEfficientNumericalMethods2009"></d-cite>. These ideas have recently been combined in the context of bilevel optimisation of neural network hyperparameters<d-cite key="pmlr-v108-lorraine20a"></d-cite>. Sambharya et al<d-cite key="sambharyaLearningWarmstartFixedpoint2024"></d-cite> propose using neural networks to learn the relationship between parameter values and optimal warm starts of numerical fixed-point optimisers and compare this learned warm start approach with a "nearest-neighbour" warm start that is similar to our method. The learned warm start performs better for problems where the problem to be solved is far away from its nearest neighbour, so that the neighbour's parameter and root combination and their gradients are uninformative. In contrast, embedded root-finding problems on adjacent steps of a simulated Hamiltonian trajectory are usually close, and very many warm starts are required for a full MCMC run, motivating our consideration of cheaper linear warm starting procedures.
>
> **Failed runs**
>
> It is correct that *guess-previous* failed on multiple benchmarks. This was also the case for the other two heuristics and is to be expected as we were aiming to explore performance on difficult non-linear root-finding problems. If there were no failures that would be a sign that we did not choose difficult enough problems. The samplers using dynamic guessing failed much less frequently than *guess-static*.
>
> To make this point clearer we have added a table with the precise number of failed runs for each heuristic and model.

---

### Author Response · Authors · 2026-04-22
**Mathematical exposition, presentation of results and choice of metrics**

Thanks very much for these detailed and constructive reviews. We have incorporated the reviewers' recommendations and believe the paper is now much better as a result. This comment summarises changes in mathematical exposition, our choice of metrics and the presentation of our results, which were mentioned by several reviewers.

**Mathematical exposition**

We have updated our paper with detailed mathematically precise introductions to the main topics of the problem setting, i.e. Hamiltonian Monte Carlo, numerical root-finding and the combination of these two things. We have also explained our contribution in more detail and with mathematical notation.

**Metrics**

All reviewers recommend reporting wall times to indicate the actually-realised performance of our implementation of dynamic guessing. Additionally, reviewer qAvm suggested simplifying our theoretical performance metric to simply count Newton steps, rather than calculating the ESS:step ratio.

We agree with both of these suggestions, so we added a table and figure reporting wall times and changed our theoretical metric to a simple count of Newton steps.

**Presentation of results**

We added table comparisons of main results, improved the explanation of our figures and revised our benchmark figures to show relative performance compared with static guessing, making them easier to understand by removing a visual element.

We would greatly appreciate further feedback about all of these topics.

---

### Decision · Action_Editor_gNPC · 2026-05-19

**Recommendation:** Accept as is

**Audience:**

Yes

**Audience Explanation:**

The reviewers, with whom I agree, point out that the scope of the paper is rather narrow; yet some small sub-set of readers of TMLR may find the heuristics found in the paper useful in their context.

**Claims And Evidence:**

Yes

**Claims Explanation:**

The main issue consistently pointed by the reviewers was a lack of formality in the problem definition, which was later resolved.
The claims are supported by numerical tests only, which are properly conducted.